# Advanced Breeding for Biotic Stress Resistance in Poplar

**DOI:** 10.3390/plants11152032

**Published:** 2022-08-04

**Authors:** Chiara Biselli, Lorenzo Vietto, Laura Rosso, Luigi Cattivelli, Giuseppe Nervo, Agostino Fricano

**Affiliations:** 1Council for Agricultural Research and Economics, Research Centre for Viticulture and Enology, Viale Santa Margherita 80, 52100 Arezzo, Italy; 2Council for Agricultural Research and Economics, Research Centre for Forestry and Wood, Strada Frassineto 35, 15033 Casale Monferrato, Italy; 3Council for Agricultural Research and Economics, Research Centre for Genomics and Bioinformatics, Via San Protaso 302, 29017 Fiorenzuola d’Arda, Italy

**Keywords:** poplar, climate change, breeding, biotic stress, resistance, QTL, genome editing

## Abstract

Poplar is one of the most important forest trees because of its high economic value. Thanks to the fast-growing rate, easy vegetative propagation and transformation, and availability of genomic resources, poplar has been considered the model species for forest genetics, genomics, and breeding. Being a field-growing tree, poplar is exposed to environmental threats, including biotic stresses that are becoming more intense and diffused because of global warming. Current poplar farming is mainly based on monocultures of a few elite clones and the expensive and long-term conventional breeding programmes of perennial tree species cannot face current climate-change challenges. Consequently, new tools and methods are necessary to reduce the limits of traditional breeding related to the long generation time and to discover new sources of resistance. Recent advances in genomics, marker-assisted selection, genomic prediction, and genome editing offer powerful tools to efficiently exploit the *Populus* genetic diversity and allow enabling molecular breeding to support accurate early selection, increasing the efficiency, and reducing the time and costs of poplar breeding, that, in turn, will improve our capacity to face or prevent the emergence of new diseases or pests.

## 1. Introduction

The term “poplar” identifies a group of about 30 tree species, belonging to the genus *Populus,* which are traditionally clustered into six sections known as *Populus*, *Tacamahaca*, *Leucoides*, *Turanga*, *Abaso*, and *Aigeiros* [1], although phylogenies based on nuclear rDNA and mitochondrial sequences corroborate the polyphyletic origin of some sections [2]. Comparative genomics is contributing to unravelling the evolution and classification of poplar species, showing that white poplar (*P. alba*) is genetically close to black cottonwood (*P. trichocarpa*), although they are currently included in *Populus* and *Tacamahaca* sections, respectively [3]. Unlike these clusters, a high level of sequence conservation has been detected between the European aspen (*P. tremula*) and quacking aspen (*P. tremuloides*), in agreement with their classification in the same *Populus* section [4].

In its native habitats, poplar shapes natural forest and woodlands but it is also the most intensively domesticated forest tree species [5]. The genome sequence of black cottonwood was the first one obtained for forest trees [6] and nowadays large genomic resources are available for poplar species. The easiness in transformation, clonal propagation, and the fact that the reproductive stages can be reached in a reasonable short time have fostered the use of poplar as a model species for genetic and genomic studies for forestry and tree breeding [7,8]. The most economically relevant poplar species exploited for breeding and for the generation of commercial hybrids include black poplar (*P. nigra*) and eastern cottonwood (*P. deltoides*), both belonging to *Aigeiros* section, quaking aspen (*P. tremuloides*), European aspen (*P. tremula*), and white poplar (*P. alba*), included in *Populus* section, black cottonwood (*P. trichocarpa*), and the Japanese poplar (*P. maximowiczii*), both classified in *Tacamahaca* section [1] (Table 1).

The economic relevance of poplar derives from its fundamental role in the wood value chain as it is cultivated to produce timber, plywood, veneer, industrial roundwood, pallets, paper pulp, fodder, and fuelwood, and to provide services (shelter, shade, and protection of soil, water, crop livestock, and dwelling) [9]. It is also employed for phytoremediation of heavy metals, contaminated soils, rehabilitation of fragile ecosystems (i.e., ecosystems exposed to desertification), and forest landscape restoration [10]. In 2016, there were about 9.6 million hectares devoted to poplar farming worldwide, 3.8 of which were grown for wood production, while the rest was mainly managed for environmental purposes [11]. Turkey, China, France, the Po River plain in Italy, and India are the regions with the largest poplar farming acreage and wood production worldwide [11], while in Europe, France, Italy, Spain, and Hungary account for about 0.5 out 0.61 million hectares devoted to poplar farming [9,11].

As they do for other forest trees, an ample range of fungi, bacteria, viruses, and pests threaten poplar farming, causing the reduction of plant growth and, ultimately, the quantity and quality of wood biomass. The main poplar diseases are caused by fungi, and the most damaging are represented by *Septoria musiva* (also known as *Sphaerulina musiva*), the causal agent of canker and leaf spot and particularly harmful in America, *Discosporium populeum*, the infectious agent of canker, which determines considerable damages in European and American areas with continental climatic conditions, the widespread poplar rust agents belonging to *Melampsora* genus, the leaf spot-associated *Marsonnina brunnea*, and the spring leaf and shoot blight *Venturia* species, widely diffused in the northern hemisphere [12,13,14,15,16,17]. An exhaustive summary and description of fungal species that damage poplar farming along with other pathogens and pests is reported in Table 2.

Among insects, *Cryptorhynchus lapathi* represents the most damaging pest for young plantations and nurseries, *Saperda carcharias* is the most dangerous in the Mediterranean basin, *Phloemyzus passerinii*, also known as Woolly Poplar Aphid (WPA), causes up to 10% of yield losses in European and American countries, *Anaplophora glabripennis* is responsible for the destruction of hectares of Chinese poplar (*P. simonii*) stands, while, in the last years, the invasive brown marmorated stink bug (*Halyomorpha halys*) has become one of the most serious economic pests for different plant species worldwide. Other insects that have the potential to threaten poplar farming are *Cossus cossus*, *Megaplatypus mutatus*, *Agrilus suvorovi*, *Melanophila picta*, *Paranthrene tabaniformis*, and *Gypsonoma aceriana* [7,16,18,19,20] (Table 2).

Beyond fungi and insects, bacteria classified in the *Erwinia* genus, *Xanthomonas populi*, and *Lonsdalea populi* cause poplar canker resulting in consistent losses of wood biomass yield. Lastly, leaf viral pathogens (i.e., poplar mosaic virus, poplar decline virus, and arabis mosaic virus) also represent serious threats [15,16,17,21,22,23] (Table 2).

For woody plants, including poplar, biotic stresses are the results of complex biological interactions among hosts, pests and pathogens, and environmental conditions [24]. In both natural populations and specialized poplar plantations, these concurrent combinations of factors may cause a high rate of tree mortality [25]. Climate change is altering the bioclimatic factors that characterize poplar farming areas, and, in general, the growing areas of perennials and agricultural lands, and is expected to enhance the vulnerability to environmental stresses with a reduction of productivity [26]. In crops, several observations indicate that climate-warming drives the movement of pests and diseases poleward [27]. Similarly, current evidence points out that climate change underlies the expansion of the geographic ranges of diseases, affecting forest species towards higher latitudes and the emergence of pathogens in new territories where susceptible plantations are located [28,29]. In poplar, simulation models predict an increase of the average degree of susceptibility to *Melampsora larici-populina* for the next decades (2031–2070) in New Zealand [30]. Under the new climate scenarios, the understanding of the relationship between changing environmental conditions and the spread of poplar pests and pathogens is pivotal to drive breeding activities and to adopt mitigating policies to tackle or anticipate the emergences of diseases in new territories.

Traditional methods for the prevention of poplar pests and diseases are based on the selection of clones well adapted to soil (pH, salinity, and calcium content) and climatic conditions with some resistance to drought. Well adapted clones can avoid abiotic stresses that can foster the incidence of biotic attacks. Good clonal adaptability coupled to the choice of proper cultural practices (i.e., low plant density, polyclonality, fertilization, and irrigation) and the use of resistant clones represent prevention systems that are commonly adopted in several plantation areas.

The recent advances in genomics, quantitative trait loci (QTLs) mapping, genomic prediction (GP), and genome editing (GE) have the potential to accelerate and improve the traditional poplar breeding, and more in general tree breeding, to cope with old and emerging pests and diseases [31]. In this review, we firstly revised the existing diversity for biotic stress resistance in untapped poplar germplasms and its relevance to support improvement efforts to create new clones resistant to biotic stresses. Secondly, we examined the most recent advancements in genetics, genomics, and biotechnology that are currently used, or might be used, for improving the selection of new poplar clones, highlighting how genomic tools could be deployed for ameliorating and fastening poplar breeding for biotic stress resistance.

## 2. Harnessing Poplar Diversity to Improve Pest and Disease Resistance

The narrow genetic diversity of cultivated commercial clones of poplar and the intensive poplar farming adopted worldwide are exacerbating the need of new resistant genotypes. Unlike spontaneous forests, poplar farming is largely based on a few F_1_ interspecific clones that have been selected to maximize wood production in specific environments [32]. Moreover, several commercial clones do not often offer appropriate resistance levels to cope with the emergence of pests and diseases moving in new territories as consequence of climate change. This is especially true in the Mediterranean area where a single clone, the Italian selection ‘I-214’, is the most used particularly in monoclonal stands [16]. For instance, leaf rust can cause up to 60% of yield losses in poplar stands [33], while, as mentioned above, up to 10% reduction of potential production has been reported for WPA [7].

To manage the attack of some of these pathogens and pests (i.e., leaf rust, leaf spot, shoot blight, bacterial canker, and WPA) several breeding programmes targeting biotic stress resistance have been developed worldwide [34,35,36,37,38,39,40,41,42]. All these conventional breeding programmes rely on the availability of germplasm diversity as a source of donor genotypes carrying resistant alleles, which are pivotal to succeed in developing high-performance clones. Consequently, the identification of new sources of resistance to biotic stresses in untapped poplar germplasms and the breeding of a new generation of clones represent the main long-term strategy for coping with the effects of climate change and contrasting the spread of diseases. 

Several studies have assessed the existence of genetic diversity for biotic resistance and there is evidence that individuals sampled in natural forests can provide new alleles at resistance loci [43]. The analysis of trait diversity carried out in 47 genotypes of *P. nigra* collected in several location in Europe has shown various levels of resistance against *S. musiva* [43]. The same pattern has been substantiated in eastern cottonwood (*P. deltoides*), for which a correlation between genomic diversity and resistance to different lineages of *S. musiva* has been highlighted [44]. Following the same research line, it has been shown that trees sampled from wild populations of black poplar (*P. nigra*) exhibit genetic variation for resistance against *M. larici-populina* [45], and the identification of two highly resistant trees corroborates the finding that untapped genetic materials sampled in natural forests might be relevant sources of resistance to sustain poplar breeding programmes [46]. 

While large studies have pointed out diversity for resistance to biotic stresses, the bioclimatic parameters that characterize the geographic origin of poplar germplasm might be useful to carry out initial screening as there is a high chance to identify resistant trees in areas subjected to the natural occurrence of pathogens. For instance, eastern cottonwood and black cottonwood (*P. trichocarpa*) trees originating from more humid environments exhibit, in general, higher levels of leaf rust resistance, compared to trees sampled in drier environments, which have evolved under weaker pathogen selection intensity [47]. This approach has been largely substantiated for cereal crops [48] and might be used to mine large poplar collections and assemble diversity panels to improve biotic stress resistance.

The effective exploitation of untapped poplar germplasms requires field trials and rapid assays to discover and evaluate resistant alleles effective against different pathogen races [34]. Typically, these assays include tests carried out in controlled conditions (i.e., greenhouses or growth chambers) or in laboratories. Moreover, the reliability of these assays should depend as less as possible on testing conditions or tree age and dimensions. In poplar, the expression of resistance against fungal pathogens often occurs during juvenile stages and has fostered the development of phenotyping methodologies in controlled conditions, facilitating breeders’ work [49]. Assays for evaluating resistance against *M. brunnea* and *X. populi* have been developed and, currently, allow early and accurate selection of resistant poplars at juvenile stages [49]. A better control of the environmental factors in the juvenile evaluation improves the efficiency of early selection for *Melampsora* species [49], for which the resistance is often evaluated on one-year-old plants in controlled conditions [50]. Other assays for testing the resistance to *M. larici-populina* in laboratory conditions are based on the scoring for the appearance or absence of fungal spores in poplar leaf disks upon artificial inoculation [51]. Interestingly, the estimation of *V. populina* resistance in controlled conditions has been also validated using randomized clonal trials organized in sites where this pathogen severely attacks poplar plantations [52].

Breeding for insect resistance exacerbates the need of solid phenotyping methodologies and testing assays as insects are known to exhibit preferences towards certain individuals in monoclonal plantations [34], hampering the identification of resistant genotypes. Nevertheless, at least for three different species (WPA, *G. aceriana*, and *S. salicis*), methods for assessing resistances in controlled conditions have been developed [34], and, particularly, the test for evaluating the susceptibility to WPA is currently used for *P. ×canadiensis* breeding programmes and for genetic studies [7,34,49].

To fully harness advantageous alleles, poplar germplasms exhibiting the relevant biotic resistance must enter in actual breeding programmes, which are mainly based on hybrid breeding to systematically exploit heterosis [8,36,49,53]. In Europe, the most successful poplar clones belong to *P. ×canadiensis*, which derives from enforced crossing between eastern cottonwood and black poplar, *P. ×wettsteinii* originated by inter-specific crosses between European aspen and quacking aspen, and *P. ×tomentosa*, obtained by crossing white poplar and European aspen (Table 3).

The exploitation of heterosis in hybrid breeding usually requires the development of heterotic groups, including lines that may or may not be genetically related and that can exhibit heterosis in F_1_ generation. In the context of poplar breeding, heterotic groups coincide with the different sets of parental lines belonging to different species, making F_1_ inter-specific hybrids the most effective strategy to maximize heterosis [6,34,52,54]. Using both field trials and assays in controlled conditions, the public breeding programme of poplar carried out in Italy has applied semi-reciprocal recurrent selection to develop several groups of parental lines of eastern cottonwood and black poplar, which show improved resistance to WPA for eastern cottonwood and resistance to WPA, *Marssonina*, *Venturia*, and *Melampsora* species for black poplar [8,36,53].

## 3. Genome Sequences Are Pivotal to Improve Poplar Breeding for Biotic Stress Resistance 

The genome sequence of black cottonwood is a landmark for plant science as it was the first sequenced genome of a perennial tree species [6]. The assembly statistics shows that this reference sequence spans about 550 Mb, is organized in 19 chromosomes, and, as for other outcrossing species, is highly heterozygous [6,55] (Table 4).

Interestingly, the analysis of black cottonwood genome sequence shows the signature of several whole-genome duplications along with shorter tandem duplications and about 8000 paralogs [6].

After the release of this landmark genomic resource, other high-quality reference sequences have been recently assembled for eastern cottonwood clone W94 (https://phytozome-next.jgi.doe.gov/info/PdeltoidesWV94_v2_1 (accessed on 22 March 2022)), and for cultivar I-69, which shows 97.4% of anchored sequences and only 0.08% of gaps (Table 4) [59]. Additional reference sequences have been assembled and annotated for *P. pruinose* [54], desert poplar (*P. euphratica*) [55,56], the two aspen species *P. tremula* and *P. tremuloides* [4], white poplar [3,57], *P. simonii* [58], and the hybrid *P. alba × P. tremula* var. *glandulosa* 84K [60,61] (Table 4).

Overall, more than 45,000 protein-coding genes were annotated on poplar genomes and their functional characterization represents a powerful tool to help dissecting the genetic bases of biotic stress resistance and accelerating poplar breeding [4,59]. For instance, the gene ontology (GO) classification carried out on European aspen and quacking aspen genome sequences allowed the identification of genes putatively involved in disease resistance [4]. In a similar way, the genome sequence of white poplar revealed a loss of disease-resistance genes [3].

A deeper assessment of the genetic diversity in poplar has been achieved through the analysis of pan-genome, which has allowed to detect structural variants, including insertion/deletions (INDELs) and copy number variations (CNVs), in closely related poplar species (*P. nigra*, *P. deltoides*, and *P. trichocarpa*) [62]. Interestingly, the GO classification carried out on genes exhibiting copy number variants shows that many of them are implicated in resistance to stresses and diseases [62].

Poplar genomic assemblies have enabled comparative in silico analyses between unrelated species, demonstrating great potential for unravelling the functional role of genes. By comparing the genome sequences of poplar and the model plant *Arabidopsis thaliana*, several candidate genes associated to disease resistance were identified [14]. For instance, the genome assembly of black cottonwood was mined using the sequences of 15 *Arabidopsis* MLO proteins, associated to susceptibility to powdery mildew, allowing the mapping of 26 *MLO* genes on 14 black cottonwood chromosomes. On the base of *Arabidopsis*-poplar phylogeny, four of these genes were considered as potential candidates involved in poplar-powdery mildew resistance, paving the way for future studies aimed at managing the spread of this disease in poplar plantations [63]. Following the same approach, a genome-wide analysis of WRKY transcription factors encoding genes on poplar genomes led to the identification of 100 unique genes, 61 of which displaying modulated expression in response to biotic stresses [64]. In addition, Hidden Markov Models (HMMs) were applied for a comprehensive genome-wide analysis of pentatricoptide repeat (PPR) genes on black cottonwood genome and 626 of such genes were discovered, 154 of which are modulated in response to abiotic stresses and *M. brunnea* attack [65].

Beyond comparative analyses, genome sequences are enabling the identification and mapping of molecular markers, which, in turn, contribute to accelerating QTL mapping and Genome Wide Association Scan (GWAS) applications for the identification of loci underlying biotic stress resistance. The analysis of the genetic diversity carried out in a panel of 1038 black cottonwood trees, mostly sampled across their natural range, identified more than 7.4 M genetic variants including rare single nucleotide polymorphisms (SNPs) and INDELs [66]. Similarly, targeted resequencing of 579 eastern cottonwood individuals, sampled across populations spanning 15 states in the USA, allowed to detect more than 500k SNPs [67].

Exploiting available genome sequences, three SNP arrays were developed for high-throughput genotyping of poplar: the Illumina iSelect Infinium 34K array interrogates 34,131 SNPs derived from the resequencing of 34 wild black cottonwood accessions and located within or close to 3543 genes [68]; the 12K Infinium array was designed using SNPs identified by resequencing 51 black poplar trees and includes markers within or close to candidate genes for rust resistance, wood properties, water-use efficiency, and bud phenology [69]; and, lastly, a new Affimetrix multispecies SNP array (4TREE array) has been developed within the EU-funded H2020 B4EST project (https://b4est.eu/ (accessed on 23 March 2022)) and includes 13,400 poplar SNPs obtained from the sequencing data of 90 black poplar genotypes and 30 eastern cottonwood trees, respectively (data not published). Although the availability of poplar SNP arrays offers unprecedented possibilities for enabling genetic studies and improving poplar breeding, arrays are optimized for the species used for their development and cover only a fraction of the existing genetic variability within a species [70]. Recently, exome sequencing and protocols based on reduced-representation sequencing, such as Genotyping-by-Sequencing (GbS) or single-primer enrichment technology (SPET), have been used to fingerprint poplar populations and overcome the limitations of SNP ascertainment bias in genotyping arrays [7,67,71,72].

## 4. An Overview of Known Loci Co-Segregating with Biotic Stress Resistance in Poplar

The most studied poplar disease is leaf rust and *MXC3* is one of the first QTLs co-segregating with the resistance against the leaf rust agent *Melampsora × columbiana* and was mapped in a high-resolution local genetic map using Bulk Segregant Analysis (BSA) and 19 Amplified Fragment Length Polymorphism (AFLP) markers [13]. The sequences of these markers were used to screen a bacterial artificial chromosome (BAC) library of black cottonwood and positive BAC clones were subsequently assembled in a large contig, which was further extended using chromosome walking. The resulting physical map was then integrated with a high-resolution genetic map but, owing to the low recombination frequency, *MXC3* was not positionally cloned [13] (Table 5).

A second QTL for resistance against the leaf rust caused by *M. larici-populina* is the *MER* locus, which was shown to co-segregate with 11 AFLPs. These markers allowed the identification of 17 recombinants from 512 progenies of three interspecific crosses between eastern cottonwood and black poplar, using the same eastern cottonwood resistant parent, and a high-resolution local genetic map, covering 3.4 cM including the *MER* locus, was obtained. The sequencing of the AFLPs revealed similarities with nucleotide binding sites leucine-rich repeat (NBS-LRR) Resistance ® genes for three of them. Like for *MXC3*, *MER* was not positionally cloned (Table 5) [80].

Using a two-way pseudo-testcross mapping strategy, a genetic map highly saturated with AFLP and Simple Sequence Repeat (SSR) markers was constructed from 171 individuals of a F_1_ population, derived from the cross between one hybrid clone (*P. trichocarpa*, 93–968 × *P. deltoides*, ILL-101) and one eastern cottonwood genotype (*P. deltoides* clone D109). This resource was used to map both *MXC3* and *MER* loci and markers co-segregating with these loci were subsequently used to screen and assembly shot-gun sequence data form draft *Populus* genome scaffolds, generated in the pre-genomic era, to identify candidate genes for *MXC3* [73] (Table 5 and Figure 1).

A qualitative resistance locus (*R*_1_), effective towards four leaf rust strains, and nine leaf rust resistance QTLs, two of which (including *R_US_*) with broad-spectrum effects, have been mapped deploying 389 markers and a F_1_ segregating population of 343 individuals, derived from an interspecific cross of *P. deltoides × P. trichocarpa* [81] (Table 5). Fine-mapping of *R*_1_ and *R_US_*, conducted enlarging the interspecific population of *P. deltoides × P. trichocarpa* populations [81] to 1415 individuals, genotyped by fragment-based molecular markers linked to resistance, localized the two loci on chromosome 19 (Table 5). Physical maps for the two dominant and recessive *R_US_* haplotypes were generated from the *P. trichocarpa* parent, heterozygote at the locus. The genetic and physical maps were then anchored to the genome sequence, discovering clusters of NBS-LRR and serine-threonine kinasis genes for *R*_1_, and NBS-LRR genes for *R_US_* [82] (Table 5 and Figure 1). This latter example shows how the availability of poplar genome assemblies has drastically improved QTL mapping, leading to the anchoring on chromosomes and fostering the identification of candidate genes inside the confidence interval of QTLs, without the need of tedious and expensive positional cloning or physical mapping. 

Besides leaf rust, candidate genes for resistance to other pests were identified combining mapping and genomic data. Two major leaf spot resistance QTLs were mapped using a *P. deltoides* F_1_ population of 84 individuals, genotyped with 1398 AFLPs and 72 SSRs, and the sequences of molecular markers in linkage with this trait were aligned to poplar genome [76] (Table 5 and Figure 1). 

To identify common pathways of response to insect attack, a linkage map was built from 350 individuals of an interspecific cross, genotyped using fragment-based molecular markers, and assessed for the response to seven categories of insects causing leaf damage at two time points [74]. Fourteen genomic regions on nine linkage groups (LGs) correlated with plant/insect interaction and a three-step approach to combine QTL mapping and genomic information was then applied to find out candidate genes and metabolic mechanisms associated to insect response. This approach was based on: (1) searching for co-location with genes implicated in the shikimate-phenylpropanoid pathway; (2) searching for co-location with QTLs controlling leaf traits; and (3) functional classification through GO enrichment analyses of the genes detected at the level of the QTL confidence intervals [74]. As a result, similar response to different insects was discovered and 15 genes implicated in the production of phenolic glycoside were mapped within 9 QTLs, providing new understandings of the interactions between poplar and insects to be exploited in breeding programmes [74] (Table 5 and Figure 1).

The molecular bases of WPA resistance were investigated using 131 individuals of a F_1_ segregating population derived from the cross between a WPA resistant eastern cottonwood and a susceptible black poplar. This interspecific population was genotyped through GbS and available SSRs, leading to 5667 polymorphic markers that were used to create high-resolution maps for the parental lines. One major and two minor QTLs co-segregating with WPA resistance and explaining more than 65.8% of the phenotypic variance were mapped on three different LGs and candidate genes were identified [7] (Table 5 and Figure 1). GbS was also used for fingerprinting an interspecific F_1_ population of 300 trees derived from enforced crossing of the leaf rust resistant eastern cottonwood clone I-69 and a susceptible *P. simonii* genotype. Genetic maps were created using 1222 polymorphic SNPs and 11 rust resistance QTLs were mapped on nine different chromosomes (Table 5 and Figure 1). *R* gene clusters were identified withing the confident intervals of two QTLs (Table 5) and a 611-bp deletion associated to variation in rust resistance was discovered in a *R* gene, providing a marker to develop molecular diagnostic tools for rust resistance [84].

Like for QTL mapping, GWAS have been widely applied to tree species and the low linkage disequilibrium (LD) extent (less than 300 to 1000 bp for *P. nigra* [69,88,89], from 200 to several Kbp for *P. trichocarpa* [68,90,91], 1.4 Kbp for *P. deltoides* [90], 2.6 Kbp for *P. euphratica* [92], less than 400 bp to 750 bp in *P. balsamifera* [91,93], and 200 bp in *P. tremula* [94]) often results in the identification of causal variants or quantitative trait nucleotides (QTNs), when associations are discovered [95]. For poplar-pathogen interactions, 40 SNPs within 26 unique genes associated to poplar rust severity were identified using a collection of 456 black cottonwood trees genotyped employing the Illumina iSelect Infinium 34K array [85], five of which are non-race-specific and corresponding to non-*R* genes (Table 5).

## 5. Transcriptomics for Disease Resistance in Poplar

Transcriptomics is a valuable tool to compare the response to pathogen or pest attacks in closely related species or varieties of the same species, providing a snapshot of the molecular mechanisms activated or repressed during infections. This brings to the prediction of pathways and candidate genes implicated in the response to pathogens that might be useful in breeding programmes [96]. In poplar, transcriptional changes and differentially expressed genes (DEGs) upon attacks of *M. medusae*, *M. larici-populina*, and *M. brunnea* were initially analysed using expression microarrays in *P. ×canadiensis*, *P. ×generosa*, and *P. deltoides*, respectively [97,98,99,100]. 

The introduction of Next Generation Sequencing (NGS) for studying gene expression through RNA-Seq is generating comprehensive transcriptome datasets, providing the expression level, sequence variations, and transcriptional structure (i.e., alternative splicing) of each gene [101]. RNA-Seq was applied to analyse the response to *S. musiva* of two resistant and two susceptible poplar clones and pathways implicated in the resistance (oxidation-reduction, protein fate, secondary metabolism, and accumulation of defence-related gene products) or in susceptibility (hypersensitive response loci) were discovered [102]. Likewise, a comprehensive transcriptomic experiment was conducted to investigate the response to the transition from the biotrophic to the necrotrophic phase of the hemibiotrophic fungus *M. brunnea* of two susceptible poplar species, *P. deltoides* and *P. alba × P. alba* var. *pyramidalis*, at three critical time-points. Pathways activated by infections were discovered and differences were observed during the progress of infection and among the two genotypes: *P. deltoides* is more responsive to initial attack, while the main interaction between *P. alba × P. alba* var. *pyramidalis* and the fungus occurs at the necrotrophic phase. This information is potentially used to guide the development of poplar resistant varieties to leaf spot [103].

The transcriptome of eight different poplar clones belonging to three different species was analysed in response to the attack of herbivory insects and allowed the identification of common herbivory-induced genes and signalling pathways, which represent potential regulators of poplar response to insects [104].

RNA-Seq can also be directed for monitoring the expression of microRNAs (miRNAs). This application was used to study compatible and incompatible interactions between *P. szechuanica* and avirulent and virulent isolates of *M. larici-populina*: miRNAs differentially expressed in the two interactions, including miRNAs regulating disease-resistance genes, kinases, and transcription factors that, probably, contribute to mount the response to rust infection, were discovered [105].

## 6. Integration of Transcriptomic and Genomic Data

The integration of QTL mapping or GWAS with transcriptomic analyses has a great potential for sustaining breeding as it allows to better prioritize candidate genes underlying specific traits. Following this approach, leaf rust resistance loci *R*_1_, *R_US_* [81,82]_,_ and *MER* [73], previously mapped on chromosome 19, were further dissected using DEGs located on the same target genomic regions [83] (Table 5). The sequences of the molecular markers flanking these three loci were aligned to poplar reference sequence to obtain their corresponding physical positions (Table 5 and Figure 1). The orthologous sequences of eastern cottonwood, which was the donor of *R*_1_ and *MER* resistance, were assembled and candidate *R* genes were functionally annotated by BLAST searches on available databases. This analysis was integrated with an RNA-Seq experiment to compare the transcriptional response to leaf rust, at two time-points from the inoculum, studying two eastern cottonwood genotypes displaying opposite behaviour to infection (highly resistant T-120 and highly susceptible D-896). Three candidate *R* genes (one located on *R_US_* and two inside *MER*) were identified among the DEGs mapped within the target loci (Table 5) and were analysed for sequence variations in the contrasting genotypes in order to characterize haplotypic variants that could represent tools for marker-assisted selection (MAS, the use of molecular markers to select plants with better performance) [83].

Like for leaf rust resistance, a GWAS aimed at identifying genes underlying resistance to *S. musiva* was carried out using a panel of 1000 black cottonwood trees and allowed the detection of nine associations, six of which located at the level of genes encoding for receptors. To better analyse the putative role of these genes in the fungal response, RNAs from resistant and susceptible genotypes, collected at three time-points after infection, were sequenced and DEGs located within the associated loci and showing expression patterns compatible with resistance or susceptibility were discovered [86] (Table 5). Validations by binding assays and in vivo over-expression were also performed [86].

Recently, an RNA-Seq experiment carried out on *P. tomentosa* LM50 upon inoculation with *M. brunnea*, at three time-points, was integrated with a multigene association analysis conducted on a panel of 435 unrelated *P. tomentosa* individuals. This germplasm collection was high-throughput genotyped using 29,399 SNPs obtained from direct sequencing and filtered for the presence on the DEGs discovered by the RNA-Seq experiment as being implicated in *M. brunnea* response. Following this approach, it was possible to find out many key genes implicated in poplar response to *M. brunnea* with potential roles in regulating photosynthesis and plant growth, highlighting the genetic interaction between different pathways during pathogen attack [106].

A different approach is based on the conjugation between GWAS and the identification of expression-QTL (eQTLs). eQTLs are DNA variants that contribute to variation in the expression levels of a gene in different genotypes and can represent *cis*- or *trans*-regulatory elements responsible of differences in the expression of a phenotype [107]. They can be detected using transcriptomic techniques in association with mapping methods that calculate the linkage between variation in expression and genetic polymorphisms. Following this approach, the poplar *PtHCT2* gene, encoding for a hydroxycinnamoyl-CoA:shikimate hydroxycinnamoyl transferase 2 (Potri.018G105500), was associated to the response to biotic stresses (Table 5 and Figure 1) [103]. In more detail, a GWAS was conducted on 917 *P. trichocarpa* accessions, genotyped by resequencing, and regions associated to the accumulation of secondary metabolites, including the *PtHCT2* region, were discovered. To confirm the implication of this gene in the antioxidant response, an RNA-Seq experiment was conducted on leaves and xylem of 390 and 444 accessions, respectively. Sequencing results were used to perform correlations with three secondary metabolite abundances and the expression of nine *PtHCT* genes. Significant associations were discovered only for *PtHCT2*. An eQTL mapping was then performed using the transcript abundance of *PtHCT* genes, as the phenotypic variable, in the GWAS experiment. Interestingly, a *cis*-eQTL was discovered regulating *PtHCT2* in both leaves and xylem and two SNPs in this region affected the core of a W-box element, the binding site for WRKY transcription factors, known to be involved in defence response. This association was also confirmed by transient expression and co-expression networks, developed from the RNA-Seq data described by Muchero et al. [86] that revealed that *PtHCT2* responds to *S. musiva* [103].

## 7. Breeding for Biotic Stresses in Poplar: Future Perspective

In recent years, genetic studies have allowed to dissect the genetic bases of biotic resistances in poplar and this information has the potential to be exploited for tree breeding, particularly for implementing MAS and, consequently, marker-assisted breeding (MAB). Under certain scenarios, it was shown that MAS can be used to replace or improve truncation selection based on phenotypic values [108] and has the potential to accelerate the identification of the best individuals in a segregating population or germplasm collection, reducing the time and costs of breeding (Figure 2) [109].

The knowledge acquired in QTL mapping and GWAS shows that, in several cases, a few QTLs or QTNs with large effects underlie the genetic bases of biotic stress resistance [7]. Moreover, high-density DNA arrays and NGS, coupled with methods based on reduced-representation sequencing, have drastically cut down genotyping costs in poplar, as they have for other species [68,69,72]. Both factors are fostering the use of MAS in poplar breeding as, even though the bottleneck related to the long juvenile phases of perennial plants remains, it allows early selection of a limited number of individuals carrying the desired traits to bring to sexual maturity, thus reducing the spaces and costs for field trials [110]. While MAS offers undoubtedly advantages to accelerate selection, its application in plant breeding requires to appropriately choose and validate the QTLs or QTNs that underlie target traits, which implies testing QTLs/QTNs over different site-by-season combinations or in different backgrounds as their effects might disappear in other pedigrees [111]. Currently, a large fraction of QTLs for biotic stress resistance identified in poplar have not been yet validated, posing further challenges to their exploitation in poplar breeding.

The application of GP, which aims to estimate the breeding value of individuals using only genotypic information [112], for the improvement of poplar is still in its infancy. Nevertheless, the lesson learned from other crops and simulation studies points out that this methodology might be a game-changer for accelerating tree breeding as it allows increasing the genetic gain per unit time and improving selection accuracy [113]. While simulation studies applied to woody perennials corroborate the advantage of GP for the simultaneous improvement of thousands of minor-effect loci [114], its application for predicting biotic stress resistance in poplar might not outperform MAS as GWAS and QTL mapping studies showed that, in many cases, few loci explain a large fraction of the phenotypic variation. Large-scale simulation studies calibrated on poplar might provide evidence of advantages and limitations of GP for breeding for biotic stress resistance under different scenarios.

Alternative ways to accelerate the genetic gain in poplar breeding are based on genetic modification. Recently, genetic-engineering applications in poplar have been reviewed and include studies related to the generation of transgenic poplars resistant to diseases by expressing antibacterial and antifungal genes (i.e., the ones encoding for osmotin, glucanases, chitinases, lysozyme, and thaumatin) or insect *R* genes (*CRY* and genes encoding for proteinase inhibitors) [115]. Even though the production of transgenic plants is a faster and feasible alternative to traditional breeding, it is strongly influenced by the random insertion of the transgene that can result in inter-transformant variations in gene expression, due to the integration in genomic regions with low transcriptional activity, epigenetic control, or sequence-specific gene silencing [116]. Moreover, the use of genetically modified organisms (GMOs) is limited by environmental concerns and regulatory issues that are particularly strict for tree species [117].

The development of the new GE technologies based on CRISPR (Cluster Regularly Interspersed Short Palindromic Repeats)/Cas is revolutionizing the concept of genome modification because allows the introduction of insertions, deletions, or the substitutions of single base or short sequences at specific sites in a target genome in highly efficient, simple, and versatile way [118,119,120,121,122] (Figure 2). 

Even if GE in poplar is mainly focused for the improvement of growth in plantations and the use in the pulp and bio-refinery industries [123,124], applications to understand the mechanisms involved in poplar disease resistance have been reported: the CRISPR/Cas9-mediated knockout of two WRKY genes, *WRKY18* and *WRKY35*, contributed to unravelling their involvement in *P. tomentosa* resistance to *Melampsora* [125]; through molecular, genetic, and biochemical approaches, including inactivation by CRISPR/Cas9, the MYB115 transcription factor has been demonstrated to regulate fungal resistance in *P. tomentosa* by activating the biosynthesis of proanthocyanidins [126]. These examples highlight how CRISPR/Cas9 can offer a great contribution to the discovery and validation of genes underlying poplar resistance to pathogens and represent the most promising frontier for the fast and efficient generation of genome-edited resistant poplar plants, by strengthening endogenous defences through the replacement of native unfunctional alleles or promoter regions, leading to high level of expression of functional genes [124,127].

## 8. Conclusions: Are Current Knowledge on Poplar Sufficient to Support Advanced Breeding for Biotic Stress Resistance?

Biotic stresses represent major threats in both natural populations and plantations of poplar and forest trees in general. The extent of biotic stresses is intensifying as global warming is rapidly enlarging the distribution and incidence of pests and pathogens towards higher latitudes and their occurrence in new territories [28,29]. For this reason, the development of new tools to speed up poplar breeding for stress resistance/resilience is pivotal to contrast the spread of diseases.

One of the main issues of poplar breeding is tied to the biology of poplar that, as for other perennials, displays late expression of relevant traits and late development of reproductive structures, characteristics that prolong selection cycles and, consequently, the duration of breeding programmes. In addition, long breeding cycles should be supported by long-term financial investments. Nonetheless, market needs can change over time and the new clones developed by traditional breeding might not guarantee the expected return investments. 

The usual intensive cultivation of monoclonal plantations of a small number of elite cultivars, strongly adapted to the pedo-climatic conditions of specific territories, is often criticized by environmental associations, more favourable to greater biodiversity as in natural forests, and restrictions have been posed also on the cultivation of hybrid clones in protected areas and bio-conservation sites along rivers. In this condition, selected clones of native species, alone or in mixed clones, could represent a useful alternative to interspecific hybrids in more sustainable poplar plantations. For this reason, it is necessary targeting poplar genetics to guarantee the sustainability of wood productions in the areas where the spread and emergence of pests and pathogens are expected, also because of the higher frequency and incidence of abiotic stress conditions. It is also important to improve the annual rate of genetic gain in poplar breeding programmes using fast and precise applications and exploiting untapped sources of genetic variability, taking advantage of the large poplar genetic variation worldwide and creating large genetic panels that should be shared between research groups from different countries and tested in multiple environments. The molecular characterization of these collections, in combination to common garden phenotypic data, can contribute to the unlocking of this genetic diversity available to breeders and allows the acquisition of new knowledge (i.e., multi-trait associations, specific adaptive profiles, gene pools, and mixture of genotypes) to be integrated into breeding and deployment strategies. The development and use of more diverse genotypes and the maintenance of large well-adapted populations will result in a significant increase in frequency of potentially beneficial allelic variants.

The annotated poplar genomes and the new genotyping platforms have largely contributed to the mining of the interaction between poplar and pests or pathogens, but the current knowledge is still insufficient to face climate-change challenges. For example, because of the rapid LD decay measured for poplar, many molecular markers are needed to capture the effects of all QTLs. Moreover, strong LD between markers and causative variants that control the traits of interest is desirable to achieve high values of predictive ability in GP. The available SNP arrays, developed using candidate-genomic-region approaches [68,69], represent valuable tools for genetic studies but do not provide the sufficient marker density to uncover all the genetic determinants that underly a specific trait, especially the genetic variants with small effects on phenotype. The high-processive and cost-effective NGS platforms (i.e., Nova-Seq and Illumina HiSeq 4000) could contribute increasing the panel of molecular markers (including SNPs, INDELs, or CNVs) to be used in next-generation studies. 

The understanding of the relationship between changing environment and the spread of poplar diseases, using accurate prediction models of the effects of climate change on poplar plantations, is pivotal to define new breeding objectives based on the enlargement of the genetic backgrounds in parentals for the generation and selection of new clones with large biodiversity and adaptability to changing environments. 

Finally, the activation of new cooperation projects, involving geneticists, eco-physiologists, wood technologists, and stakeholders, as poplar growers and wood users, would lead to the definition of short- and medium-term objectives for the development of innovative genetic materials with wide variability to cope with future highly uncertain environments and for the definition of cost- and time-effective strategies that meet rapid changes even considering the ecological and economic contexts.

## Figures and Tables

**Figure 1 plants-11-02032-f001:**
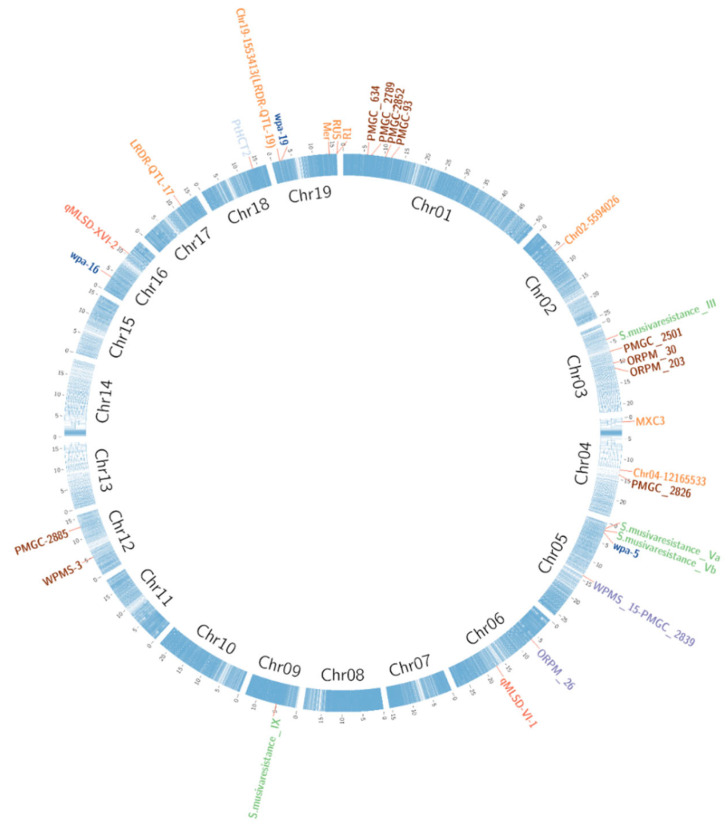
Projection of known loci co-segregating with resistance to pests and diseases on the reference sequence of *P. trichocarpa*. For *MXC3*, *MER* [73], ORPM_26, WPMS_15–PMGC_2839 [75], *qMLSD-VI-1,* and *qMLSD-XVI-2* [76], the physical position has been determined blasting the sequences of the corresponding primers on the reference genome. Loci co-segregating with the resistance to insect, leaf rust, *Sphaerulina musiva*, WPA, *Schizoempodium mesophyllincola*, *Marsonnina brunnea*, and other biotic stresses are reported in brown, orange, light green, blue, purple, red, and light blue, respectively. The 19 chromosomes of *P. trichocarpa* are coloured according to the density of annotated genes.

**Figure 2 plants-11-02032-f002:**
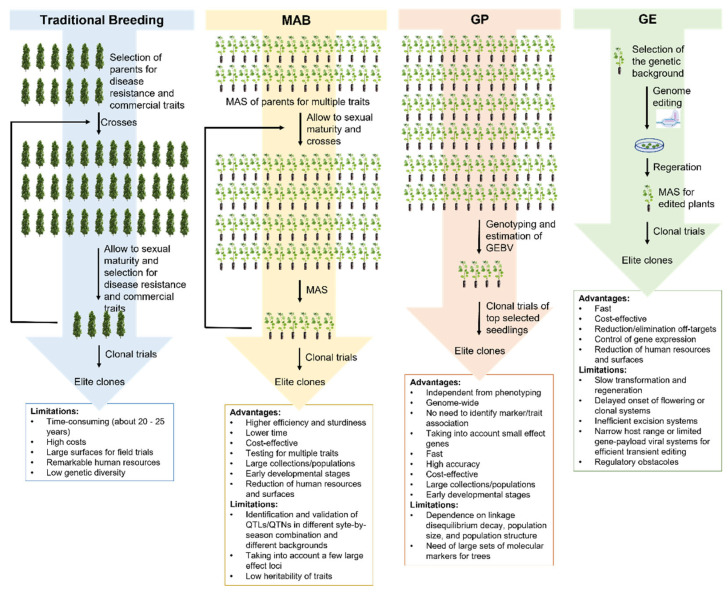
Comparison of truncation selection approaches based on phenotype values, marker-assisted breeding (MAB), genomic prediction (GP), and genome editing (GE).

**Table 1 plants-11-02032-t001:** List of poplar species exploited for breeding and for the generation of commercial hybrids.

Species	Section	Main Use
*P. deltoides*	*Aigeiros*	Female parent of *P. ×canadiensis* Parent of *P. ×generosa*
*P. nigra*	*Aigeiros*	Male parent of *P. ×canadensis*
*P. alba*	*Populus*	Parent of cross hybrids with *P. tremuloides*
*P. tremula*	*Populus*	Female parent of *P. ×canescens* Parent of cross hybrids with *P. tremuloides*
*P. tremuloides*	*Populus*	Parent of cross hybrids with *P. tremula*
*P. trichocarpa*	*Tacamahaca*	Parent of *P. ×generosa*
*P. maximowiczii*	*Tacamahaca*	Parent of cross hybrids with *P. deltoides*

**Table 2 plants-11-02032-t002:** List of the main poplar fungal, bacterial, and viral pathogens, and pests with the corresponding damages, diffusion areas, incidence of damages, and main hosts. The incidence of the pathogen has been evaluated on the base of direct field experiences, documentation on the National reports available on the IPC (International Poplar Commission) website (https://www.fao.org/ipc/en/ (accessed on 28 July 2022)), and literature searches.

Pathogen/Pest		Damages	Area	Incidence	Species
Fungi	*Alternaria alternata*	Leaf blight	India, China	Not severe	*Populus* spp.
	*Apioplagiostoma populi*	Bronze leaf	North America	Considerable	*Populus* spp.
	*Armillaria* spp.	Decline sectors of the crown, stunted vegetation	Atlantic-Mediterranean Europe, Southern Russia, Georgia, Syria, North Africa, tropical Africa, USA	Not severe	*Populus* spp.
	*Botrydiplodia populea*	Canker	China	Not severe	*Populus* spp.
	*Ceratocystis fimbriata*	Black or target canker	Alaska, USA, Quebec, Poland, India	Not severe	*P. tremuloides*, *P. deltoides* in India
	*Cercospora populina*	Leaf blotch	India	Considerable	*P. deltoides*
	*Ciborinia whetzelii*	Ink-spot disease	Canada, Northern USA	Not severe	*P. tremuloides*
	*Cladosporium humile*	Phylloptoses	India	Considerable	*P. ciliata*
	*Corticium salmonicolor*	Pink disease	India	Not severe	*P. deltoides*, *P.* ×*euramericana*, *P. yunnanensis*
	*Cryptosphaeria lignyota*	Snake canker and woody decay	Alaska, USA	Not severe	*P. tremuloides* (mainly), *P. balsamifera*, *P. thrichocarpa*, *P. deltoides*
	*Cytospora* spp.	Cytospora stem canker	Worldwide, mainly Central and Southern Italy, Eastern Europe, Near East, Northern India, and West-Central USA	Not severe, attacks occur under stresses or poor agronomic management	*Aegiros*, *Tacamahaca*, and *Leuce* sections
	*Drechslera maydis*	Leaf blight	India	Considerable	*P. deltoides*
	*Diaporthe* spp.	Phomosis stem canker	Germany, Italy, Argentina, Canada, Portugal, Japan, USA	Not severe	*Aegeiros* hybrids, *P. deltoides*, *P. alba*, *P.* ×*euramericana*, *P. nigra*, *P. maximowiczii*
	*Diplodia tumefaciens*	Bark alterations, woody galls	Northern Europe, Canada, Northern USA	Not severe	*P. tremula*, *P. tremuloides*, *P. balsamifera*
	*Discosporium populeum*	Canker	Worldwide, mainly Eurasia, North Africa, North America, and Argentina	Considerable	*Aegiros* section
	*Dothichiza populea*	Dothichiza stem canker	Europe, North America	Considerable	*Populus* spp.
	*Dothiorella gregaria*	Bark necrosis, blister canker, ulcer disease	China	Not severe	*Populus* spp.
	*Encoelia pruinosa*	Sooty-bark canker	Alaska, Western Canada, Mid-West USA, Northern Mexico, Norway	Considerable	*P. tremuloides*, *P. balsamifera*
	*Erysiphe adunca*	Powdery mildew	Italy	Not severe	*P. nigra*
	*Gibberella* spp.	Fusarium stem canker	Europe, North America	Not severe, limited to nurseries	*Aegiros* and *Tacamahaca* sections and intersectional hybrids, *P. trichocarpa*, *P.* ×*euramericana*
	*Glomerella cingulata*	Leaf and shoot blight	North-Western America, India, France	Not severe	*Populus* spp.
	*Hypoxylon mammatum*	Canker	North America, Europe	Considerable	*P. tremuloides* in North America, *Leuce* section in Europe
	*Linospora* spp.	Leaf blight	Eurasia, USA, Canada	Not severe	*P. balsamifera*, *P. deltoides*, *P. trichocarpa* ×*P. deltoides* (*L. tetraspora*), *P. alba*, *P. tremula*, *P. tremuloides*, *P. grandidentata* (*L. ceuthocarpa*)
	*Marssonnina brunnea*	Leaf spot	Worldwide	Considerable	*Populus* spp.
	*Melampsora* spp.	Leaf rust	Europe, Australia, New Zealand, Southern Africa, Argentina, North America, India, Japan, Canada	Considerable	*Populus* spp.
	*Nectaria galligena*	Canker	Central Europe, Rocky mountains	Not severe	*P. tremuloides*
	*Phaeoramularia maculicola*	Leaf spot	North America, Scandinavia, India	Not severe	*Aegiros*, *Tacamahaca*, and *Leuce* sections
	*Phoma exigua*	Cortical lesions	The Netherlands	Not severe	*P. nigra*, *P.* ×*euramericana*, *P. trichocarpa*
	*Phyllactinia guttata*	Powdery mildew	Southern Asia	Not severe	Euramerican poplars
	*Phyllosticta* spp.	Leaf spot	Europe, Argentina, Southern Australia, Japan, India	Not severe	*Populus* spp.
	*Rhizoctonia solani*	Leaf web blight	India	Considerable in nurseries and young plantations in humid conditions	*Populus* spp.
	*Rhytidiella moriformis*	Rough bark or cork bark	Canada		*P. balsamifera*, *P. tremuloides*
	*Rosellinia necatrix*	Dematophora root rot	Worldwide, mainly Italy, Portugal, Southern Africa, and India	Considerable in intensive plantations in warm- temperate or sub-tropical climates	*Populus* spp.
	*Septoria* spp.	Canker and leaf spot	North-Central Europe, North America	Considerable	*Aegiros* × *Tacamahaca*
	*Septotinia podophyllina*	Leaf blotch	North America, France, Holland, ex-Yugoslavia, ex-Czechoslovakia, Russia, Korea, Japan	Not severe	*Leuce*, *Aegiros*, and *Tacamahaca* sections
	*Sphaceloma populi*	Anthracnoses	India, Europe, Japan, Argentina	Not severe	*Aegiros* and *Tacamahaca* sections
	*Taphrina populina*	Leaf blister	Worldwide	Not severe	*Aegiros*, *Tacamahaca*, and *Leuce* sections, *P. alba*
	*Uncinula adunca*	Powdery mildew	Eurasia, North America	Not severe	*Leuce*, *Aegiros*, and *Tacamahaca* sections
	*Venturia* spp.	Spring leaf and shoot blight	Eurasia, North America, North Africa, China	Considerable	*Populus* spp.
Bacteria	*Erwinia* spp.	Bacterial twig canker	North America, Europe	Considerable	*Populus* spp.
	*Lonsdalea populi*	Bark canker	Europe, China	Considerable	*P.* ×*euramericana*
	*Phytophtora* spp.	Root rot	Europe, Africa, USA, South America, Eastern Asia, Australia, New Zealand	Not severe	*Populus* spp.
	*Pseudomonas syringae*	Bacterial blight	Worldwide	Considerable	*Populus* spp.
	*Sphingomonas* spp.	Bark canker	Worldwide	Not severe	*Populus* spp.
	*Xanthomonas populi*	Canker	Europe, North America	Considerable	*Aegiros* and *Tacamahaca* sections
Viruses	Arabis mosaic virus	Leaf mosaic	Europe, America, Japan, New Zealand	Considerable	*P.* ×*euramericana*
	Poplar decline virus	Leaf spot, necrosis	America	Considerable	*Populus* spp.
	Poplar mosaic virus	Leaf mosaic	Worldwide	Considerable	*P. nigra*, *P. deltoides*, *P. trichocarpa*, *P. candicans*, *P.* ×*euramericana*
	Potato virus Y	Mottling or yellowing of leaflets, leaf crinkling, leaf drop	Worldwide	Not severe	*P. tremuloides*, *Aegiros* section
	Tobacco necrosis virus	Vein necrosis	Worldwide	Not severe	*P. tremuloides*
	Tobacco rattle virus	Mottling, chlorotic or necrotic local lesion, ringspots or line patterns, necrosis	Worldwide	Not severe	*Populus* spp.
	Tomato black ring virus	Mottling, deformation, leaf necrosis	Worldwide	Considerable	*P. balsamifera*
Insects	*Aceria parapopuli*	Soap sucker, galls	North America	Can be considerable	*Populus* spp.
	*Agrilus suvorovi*	Borer	Europe, Asia	Not severe, more considerable on one-year plants	*P. tremula*, *P. deltoides*, *P. alba*
	*Altica populi*	Defoliation	North America	Not severe	*P. tacamahaca*, *P. tremuloides*
	*Anoplophora* spp.	Borer	China, North America, Japan, Northern India, Pakistan	Considerable, the most important pests in China (*A. nobilis* and *A. glabripennis*), not severe in Japan (*A. malasiaca*)	*Populus* spp.
	*Apriona* spp.	Borer	Northern India, Pakistan, China, Japan	Can be considerable	*Populus* spp., *P.* ×*euramericana* (*A. cinerea*)
	*Asymmetrasca decedens*	Defoliation	Mediterranean areas, India	Not severe	*Populus* spp.
	*Batocera lineolata*	Borer	Japan	Considerable	*Populus* spp.
	*Byctiscus populi*	Defoliation	Europe	Can be considerable	*P. deltoides* and Euramerican hybrids
	*Capnodis miliaris*	Borer	Syria, Turkey, Iran, Iraq, Southern Italy	Considerable in drought conditions	*Populus* spp.
	*Cerura* spp.	Defoliation	Continental Europe, United Kingdom	Not severe	*Populus* spp.
	*Choristoneura conflictana*	Defoliation	Canada, Alaska, North-Eastern and Central USA	Considerable	*P. tremuloides*, *P. deltoides*, *P. gradidentata*
	*Chrysomela* spp.	Defoliation	Europe, North America	Considerable in young plantations and nurseries (*C. populi* and *C. tremulae*)	*P. tremula* × *P. tremuloides*, *P. tremula* × *P. alba*, *P. alba*
	*Clostera* spp.	Defoliation	Europe, Siberia, Japan, Korea, China, India, Pakistan	Considerable	*P. tremula* (mainly), *P. euroamericana*, *P. euphratica*, *P. nigra*
	*Cossus cossus*	Borer	Europe, North Africa	Can be considerable	*Populus* spp.
	*Cryptorhynchus lapathi*	Borer	Europe, China, Japan, USA, Canada	Considerable in young plantations and nurseries	*Populus* spp.
	*Dasineura salicis*	Galls	Europe, North America	Not severe	*Populus* spp.
	*Epinotia solandriana*	Defoliation	Europe, North America	Not severe in Europe, considerable in Canada for *P. tremuloides*	*Populus* spp.
	*Gypsonoma* spp.	Borer, leaf mining, galls	Europe, North Africa, North America, Pakistan	Not severe, can be considerable in young plantations and nurseries	*P. deltoides* (*G. haimbachiana*), *P. euphratica* (*G. riparia*)
	*Halyomorpha halys*	Borer	China, Japan, Taiwan, USA, Europe	Considerable	*Populus* spp.
	*Hyphantria cunea*	Defoliation	North America, Canada, Central and South-Eastern Europe, Japan, Korea	Considerable	*Populus* spp.
	*Janus* spp.	Defoliation	East USA, South Canada, Central and Southern Europe	Not severe	*Populus* spp.
	*Leucoma* spp.	Defoliation	Europe, Middle East, Japan, America, China	Not severe	Mainly *P. alba*, *P. deltoides*, *P. nigra*, *P.* x *euramericana* hybrids
	*Megaplatypus mutatus*	Tunnels in stems	South America, Europe	Can be considerable	*Populus* spp.
	*Melanophila picta*	Borer	Bulgaria, Spain, Southern France, Italy, Portugal, Pakistan, Turkey	Not severe, attacks occur only under water stress, more considerable on one-year plants	*Populus* spp., mainly *P.* ×*euroamericana* and *P. euphratica* in Iraq, *P. nigra* is less vulnerable
	*Monosteira unicostata*	Defoliation	Mediterranean areas, Turkey	Can be considerable in young plantations and nurseries	*Populus* spp.
	*M* *ordwilkoja vagabunda*	Galls	North America, Canada	Not severe	*Populus* spp., *P. tremuloides* in Canada
	*Nematus* spp.	Defoliation	Europe, South Africa, North America	Not severe	*Populus* spp., *P. deltoides* in South Africa
	*Operophtera brumata*	Defoliation	Europe, Asia, British Columbia, North America	Not severe, higher damages during drought stress	*P. tremuloides*, *P. deltoides* × *P. nigra*
	*Orgyia* spp.	Defoliation	Europe, North America, Japan, Korea, China, Russia	Can be severe	*Populus* spp.
	*Paranthrene tabaniformis*	Borer	Centre and southern Europe, North Africa, Asia (mainly China, Northern India, and Pakistan), Canada, Russia, Finland	Considerable in nurseries of one-year plants	*Populus* spp., mainly *P.* ×*trichocarpa*
	*Parthenolecanium corni*	Soap sucker	Europe, North America, New Zealand	Can be considerable	*Populus* spp.
	*Phassus excrescens*	Borer	Japan, Korea	Not severe	*Populus* spp.
	*Phloeomyzus* spp.	Soap sucker	Europe, North Africa, South America, China	Considerable	*Populus* spp., higher resistance for *P. deltoides*
	*Phratora* spp.	Defoliation	Europe, North America, Russia	Considerable in the event of outbreaks, especially in nurseries and young plantations	*Populus* spp., *P. tremuloides* in North America (*P. purpurea purpurea*)
	*Phyllobius* spp.	Defoliation	Europe, Russia, Iran, Turkey, North America	Not severe	*Populus* spp.
	*Phyllocnistis* spp.	Leaf mining	Europe, Canada	Not severe	*P. nigra*, *P. deltoides* × *P. nigra*
	*Phyllonorycter* spp.	Leaf mining	North America, Europe	Not severe	*Populus* spp., *P. nigra* (Europe)
	*Phytobia* spp.	Borer	Europe	Considerable	*Populus* spp.
	*Platypus sulcatus*	Borer	South America, mainly Argentina	Considerable	*Populus* spp.
	*Polydrusus* spp.	Defoliation	Spain, France, Italy, ex-Yugoslavia, Hungary, Eastern Canada, North-Eastern USA	Not severe	*Populus* spp.
	*Popillia japonica*	Defoliation	Japan, USA, Canada, China, Europe	Considerable	*Populus* spp.
	*Porthetria* spp.	Defoliation	Northern hemisphere	Not severe	*Populus* spp., mainly *P. nigra* (*P. obfuscata*)
	*Saperda* spp.	Borer	Europe, Asia, North America	Considerable, the main poplar pest in the Mediterranean basin (*S. carcharias*)	*Populus* spp.
	*Sesia apiformis*	Borer	Europe, Middle East, Asia Minor, China, North America, Canada	Not severe	*Populus* spp.
	*Trichiocampus* spp.	Defoliation	Europe, Middle East, North America, Japan	Can be considerable	*P. deltoides*, *P. nigra* var. *italica*, *P. tremula*, *P. tremuloides*
	*Xyleborus dispar*	Borer	Europe, North Africa, North America	Can be considerable	*Populus* spp.
	*Yponomeuta rorrela*	Defoliation	Europe	Not severe	*Populus* spp., mainly *P. alba* (*Y. gigas*)
	*Zeuzera pyrina*	Borer	Central Europe, Mediterranean basin, Asia, India, Japan, North America, South Africa	Can be considerable	*Populus* spp.

**Table 3 plants-11-02032-t003:** List of the hybrid clones obtained in poplar breeding programmes.

Hybrid	Cross
*P. ×canadiensis*	*P. deltoides × P. nigra*
*P. ×generosa*	*P. trichocarpa × P. deltoides*
*P. ×tomentosa*	*P. alba × P. tremula*
*P. ×wettsteinii*	*P. tremula × P. tremuloides*
*P.* *×interamericana*	*P. deltoides × P. trichocarpa*

**Table 4 plants-11-02032-t004:** List of the poplar assembled reference sequences, including total size, coverage of sequencing, number of scaffolds, N50 scaffold and contig sizes, the percentage included in chromosomes, and the percentage of repetitive elements (retrotransposons, transposons).

Species	Total Size (Mb)	Coverage (X)	n. Scaffolds	N50 Scaffold Size (kb)	N50 Contig Size (kb)	Chromosomes (%)	Repetitive Elements (%)	Protein- Coding Genes	Non-Coding RNAs	References
*P. trichocarpa*	423	9.44	1446	19,500	552.8	84.53	48.07	42,950	817 tRNAs, 88 snRNAs, 427 snoRNAs, 169 miRNAs	[6]
*P. euphratica*	496.5	312	9673	482	40.438		44	34,279	764 tRNAs, 706 rRNAs, 4826 snRNAs, 266 miRNAs	[56]
*P. euphratica*	574.35	246.54	507	28.59	900	98.85	56.95	36,606	8767 long non-coding RNAs	[55]
*P. pruinosa*	479	125	78,960	698.5	14		45.47	35,131		[54]
*P. tremula*	390	98	216,318	42.844			21.54	35,984		[4]
*P. tremuloides*	378	86.35	164,504	15.222			22.09	36,830		[4]
*P. alba* var. *pyramidalis*	466	320	17,797	459.178	26.535		44.61	37,901	940 tRNAs, 569 rRNAs, 123 snRNAs, 1050 miRNAs	[3]
*P. alba*	416	130	1285		1180		45.16	32,963	764 tRNAs, 706 rRNAs, 4826 snRNAs, 266 miRNAs	[57]
*P. simonii*	441	138	686	194		90.2	41.47	45,459	1177 tRNAs, 290 rRNAs, 618 snRNAs, 1153 miRNAs	[58]
*P. deltoides* W94	446.8	62.94	1375	21,700	590.2	90.2		44,853		
*P. deltoides* I-69	429	273	934	21,500	2620	97.4	32.28	44,853		[59]
Poplar 84K (*P. alba × P. tremula* var. *glandulosa*)	747.5	119.79	1384	19,600	1990	94.98	24.40	85,755	1312 tRNAs, 1140 rRNAs, 1126 snRNAs, 1983 miRNAs	[60]
Poplar 84K (*P. alba* × *P. tremula* var. *glandulosa*)	781.36 (405.31 subgenome A; 376.05 subgenome G)	189	2109 (1179 subgenome A; 930 subgenome G)	3660 (5430 subgenome A; 2150 subgenome G)			43.7 subgenome A; 40.5 subgenome G	38,701 subgenome A; 38,449 subgenome G		[61]

**Table 5 plants-11-02032-t005:** Summary of poplar QTLs associated to pest and disease resistance. For each QTL, the pathogen, the genetic resources used for the identification, the name, the LG/chromosome, the physical or genetic position, the peak marker/markers, and the candidate genes are reported when indicated in the corresponding reference/references. For *MXC3* and *MER* [73], the QTLs associated to the response to insects [74], ORPM_26, and WPMS_15–PMGC_2839 [75], and *qMLSD-VI-1* and *qMLSD-XVI-2* [76] the physical positions have been determined by blasting the sequences of the corresponding primers on *P. trichocarpa* genome (https://phytozome-next.jgi.doe.gov/info/Ptrichocarpa_v4_1 (accessed on 21 March 2022)).

Pathogen	Genotypes	QTL/Locus Name	Markers	LG/Chr./Sc.	bp	cM	Markers	Candidate Genes	Reference
*Septoria* *populicola*	F_2_ 331 (107 individuals): *P. trichocarpa* (93-968, R) × *P. deltoides* (ILL-129, S)	FLD94	P1064-B15_17	LG X			RFLP, RAPD, STS-343		[77]
FLD94	P13292-P1043	LG M			
FLD95	P1064-B15_17	LG X			
FLD95	P1322-P1310	LG A			
*Melampsora* *medusae*	F_2_ 331 (107 individuals): *P trichocarpa* (93-968, R) × *P. deltoides* (ILL-129, S)	Mmd1	P222	LG Q		5.1 cM from P222	RFLP, RAPD, STS-343		[78]
*Melampsora* *medusae*	F_1_ C9425DD (116 individuals): *P. deltoides* (7300501, S) × *P. deltoides* (7302801, R)	Lrd1	OPG10_340_OPZ19_1800_	2.6 cM from OPG10_340_7.4 cM from OPZ19_1800_			RAPD-84		[79]
*Melampsora* *Xcolumbiana*	F1 545 (1902 individuals): *P. trichocarpa* (clone 383-2499, R) × *P. deltoides* (clone 14-101, S)	MXC3	CGA.TCT_01GAC.TAC_01			0.68-2.05	AFLP–19 linked		[13]
*Melampsora* *larici-populina*	F1 87001 (139 individuals) and 95001 (77 individuals): *P. deltoides* (S9-2, R) × *P. nigra* (Ghoy, S); F_1_ 87002 (106 individuals) and 95002 (120 individuals): *P. deltoides* (S9-2, R) × *P. trichocarpa* (V24, S); backcross 95003 (70 individuals): (*P. deltoides* (S9-2) × *P. nigra* (Ghoy), R) × *P. nigra* (Ghoy, S)	MER	E40G37E39G01E44G09E32G43E45G29E47G14E39F39rE48G14E61G36E51G05E43G28			3.4 (interval)	AFLP–11 linked	AF393736_NBS/LRRAF393738_NBS/LRRAF393739_NBS/LRR	[80]
*Melampsora* *larici-populina*	F_1_ 13 (171 individuals): clone 52-225 (*P. trichocarpa* 93-968 × *P. deltoides* ILL-101, R) × *P. deltoides* (D109, S)	MXC3	STS1_A, STS3, O_349, O_356	LG IV; Chr. 4	580,744-713,007	4.1	SSR, STS, AFLP-588	NP_195325.1_thaumatin NP_173432.2_thaumatin NP_197963.1_disease-resistance protein NP_177296.1_disease-resistance protein(LRR) T10504_disease-resistance protein Cf-2.1 T10504 disease-resistance protein Cf-2.1	[73]
MER	T4_3, S2_19, R_7, O_206, S1_8	LG XIX; Chr. 19	13,586,903(O_206)	33.6	
*Melampsora* *larici-populina*	F_1_ (343 individuals): *P. deltoides* (73028-62, R) × *P. trichocarpa* (101-74, S)		E4M1-4	LG TXI		0	AFLP, RAPD, SSR, SNP, RFLP, phenotypic markers-391		[81]
	E2M6-42	LG DIII		24.4	
	E2M4-16	LG D?		0	
	R_us_	LG T?		10.2	
	R_us_	LG T?		11.1	
	R_us_	LG T?		13.1	
	R_us_	LG T?		15.1	
	R_us_	LG T?		17.1	
	R_us_	LG T?			
	E5M5-4	LG TXII			
	E5M5-7	LG T?		6.0	
	E5M5-7	LG T?		8.2	
	E5M5-7	LG T?		6.0	
	E5M5-7	LG T?		8.0	
	E4M4-10	LG DVI		158.4	
	rE1M4-8	LG TXII		66.3	
	E4M4-10	LG DVI		168.4	
	rE2M4-10	LG DIII		141.4	
	rORPM277	LG DXIX		133.0	
	rORPM277	LG DXIX		137.0	
	R1	LG DXIX		144.0	
	R1	LG DXIX		145.0	
	E1M2-8	LG DXIX		117.0	
	E1M2-8	LG DXIX		125.0	
*Melampsora* *larici-populina*	F_1_ (1415 individuals): *P. deltoides* (73028-62, R) × *P. trichocarpa* (101-74, S)	R1	G_79–I_920-3	Chr.19	16,965,396–17,119,994	3.90–4.00	SSR, STS, AFLP, RAPD-68	BED-NBS-LRRTIR-NBS-LRRSerine threonine kinase	[82]
Rus	Is_165_1–RGAs297	Chr.19		5.50–6.00	TIR-NBS-LRR
*Melampsora* *larici-populina*		R1	I_1211–I_920_3	Chr.19	16,965,396–17,119,994		Fragment-based		[83]
Rus	14N08-F–RGAs135-1	Chr.19	16,441,457–16,460,757		EVM0026813_TNL
Mer	O_263–O_206	Chr.19	13,586,903–15,058,693		EVM0004305_CNLEVM0025825_STK
*Melampsora* *larici-populina*	F_1_ (300 individuals): *P. deltoides* (I-69, R) × *P. simonii* (XYY, S)	201709ab	Chr02-5594026	LG 2, Chr.2	5,594,026	70.49; 65.49–75.49	SNP–1222		[84]
201707ab	Chr04-12165533	LG 4, Chr.4	12,165,533	76.57; 71.57–81.57	
201709p2		LG 8, Chr.8		191.79; 78.00–210.00	
201809p2	Chr14-18570439	LG 9, Chr.9		0.00; 0.00–2.00	
201809p2		LG 10, Chr.10		228.24; 226.58–252.00	
201709p2		LG 12, Chr.12		146.0; 142.00–150.00	
201809p2		LG 13, Chr.13		70.00; 65.00–75.00	
LRDR-QTL-17 (overlapped region of 201709p2 and 201809p2)	Chr17-11257300-Chr17-12346306	LG 17, Chr.17	11,257,300–12,346,306	65.67; 64.00–68.00; 66.00; 62.72–72.00	



Potri.017G104100 15 disease-resistance genes
LRDR-QTL-19 (overlapped region of 201707p2 and 201809p2)	Chr19-1553413	LG 19, Chr.19	1,553,413	23.19; 18.19–28.19; 21.19; 19.19–26.78	21 disease-resistance genes
*Melamspora* *xcolumbiana*	Collection *P. trichocarpa* (456 individuals)		23949327	Sc.5	23,949,327		SNP–34K	IQD32	[85]
	1402770	Sc.6	1,402,770		FAR1
	8261867	Sc.8	8,261,867		PIP5K
	19215715	Sc.10	19,215,715		PRR7
	2955	Sc.143	2955		NRT2.4
Insects	F_2_ 331 (350 individuals): *P. trichocarpa* (93-968) × *P. deltoides* (ILL-129)			LG Vb; Chr. 5		4; 0–19	SSR, AFLP-183		[74]
		LG I; Chr.1	5,467,692(PMGC_634); 6,435,691(PMGC_2789)	9; 0–24	4 PG genes
		LG XIV; Chr. 14		0; 0–28	1 PG gene
		LG III; Chr.3	9,528,665(ORPM_30); 10,796,665(ORPM_203)	37; 29–46	
		LG IV; Chr.4	13,371,978(PMGC_2826)	59; 45–85	1 PG gene
		LG Va; Chr. 5		76; 62–86	1 PG gene
		LG XVII; Chr. 17		54; 35–69	2 PG genes
		LG Va; Chr. 5		19; 0–41	1 PG gene
		LG VIIIa; Chr. 8		27; 12–27	
		LG XVII; Chr. 17		50; 33–70	2 PG genes
		LG I; Chr.1	9,764,020(PMGC_2852), 11,239,328(PMGC_93)	74; 32–125	2 PG genes
		LG VI; Chr. 6		144; 134–144	
		LG XII; Chr. 12	4,407,861(WPMS_3), 12,292,045(PMGC_2885)	17; 0–24	1 PG gene
		LG III; Chr.3	6,609,278(PMGC_2501)	14; 0–31	
*Phloeomyzus**passerinii* L.	F_1_ (131 individuals): *P. deltoides* (D0-092b, R) × *P. nigra* (N074, S)	wpa-5	5_2426240	LG V, Chr.5	1,975,251–2,578,834	43.7	SNP, SSR–5667	NPK1-related protein kinase 1	[7]
wpa-16	16_3345538, 16_3345877	LG XVI, Chr.16	2,980,973–3,749,017	43.4	CCCH-type zinc finger protein with ARM repeat domain
wpa-19	78_83250, 78_83287, 78_83295	LG XIX, Chr.19	2,071,803–3,238,172	44.8	14 TIR-NB-LRR disease-resistance genes Phospholipase A2
*Schizoempodium mesophyllincola*	F_2_ 331 (376 individuals): *P. trichocarpa* (93–968, R) × *P. deltoides* (ILL-129, S)		ORPM_26	LGIII; Chr.6	6,013,759–6,013,972	33.642–59.393	AFLP, RAPD, RFLP, SSR-841		[75]
	PMGC_2889B	LGI; Chr.1		108.686–118.167	
	WPMS_15-PMGC_2839	LGV; Chr. 5	23,655,307–25,782,064	63.447–76.062	
*Sphaerulina* *musiva*	Collection *P. trichocarpa* (1081 individuals)			Chr.3	3,517,268		SNP–8,253,066	Potri.003G028200_RLP	[86]
		Chr.5	942,545		Potri.005G012100_RLP
		Chr.5	1,440,266		Potri.005G018000_G-type lecRLK
		Chr.9	4,548,711		Potri.009G036300_L-type lecRLK
Biotic stress,*Sphaerulina* *musiva* responsive	Collection *P. trichocarpa* (917 individuals)	*PtHCT2*	Chr18:13249087	Chr. 18	13,222,67–13,252,693		SNP–8,253,066	Potri.018G105500	[86,87]
*Marsonnina* *brunnea*	F_1_ (84 individuals): *P. deltoides* (Zhongshi-8, R) × *P. deltoides* (D-124, S)	qMLSD-VI-1	P_2217-G_2034	LG VI, Chr.6	16,592,305–17,904,816	118.2; 92.2–137.8	SSR, AFLP–1398	Potri.006G164600.1Potri.006G171300.1 Potri.006G166700.1Potri.006G166800.1	[76]
qMLSD-XVI-2	P_2143–P_204	LG XVI, Chr.16	10,022,916–12,773,381	138.3; 128.3–144.6	Potri.016G115800.1Potri.016G115900.1 Potri.016G116000.1Potri.016G116100.1 Potri.016G114400.1Potri.016G107200.1 Potri.016G109200.1Potri.016G122700.1 Potri.016G123300.1Potri.016G123500.1

## Data Availability

Not applicable.

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
