# Peer review of "Advanced Breeding for Biotic Stress Resistance in Poplar"

_plants, 2022, doi:10.3390/plants11152032_

Round 1

Reviewer 1 Report

1.       The author should add the current issues in poplar breeding.

2.       In Table 2, the author listed the main damages in poplar caused by fungal, bacterial, viral, pathogens, and pests.

Please add the traditional methods for prevention of pests and diseases, so it can be contrasted with the new breeding methods mentioned below. In addition, please make an explanation how you can evaluate “INCIDENCE” is not severe or considerable.

3.       In Figure 2, please add the disadvantages of MAB, GP and GE breeding.

Author Response

Authors would like to thank the Reviewers for the critical assessment of the manuscript.

All Reviewer's comments have been considered and the manuscript has been modified according to the suggestions. Authors’ replies to each observation from the Reviewer are reported point-by-point below. 

REVIEWER 1

  1. The author should add the current issues in poplar breeding.

Authors’ reply: Conclusions have been implemented by describing the current issues in poplar breeding, following reviewer’s suggestion.

  1. In Table 2, the author listed the main damages in poplar caused by fungal, bacterial, viral, pathogens, and pests.

Please add the traditional methods for prevention of pests and diseases, so it can be contrasted with the new breeding methods mentioned below. In addition, please make an explanation how you can evaluate “INCIDENCE” is not severe or considerable.

Authors’ replies: Traditional methods for the prevention of pests and disease in poplar plantations have been described in Introduction from lane 111 to lane 117, as suggested by the reviewer.

An explanation of the evaluation of the “INCIDENCE” indicated in Table 2 has been described in the caption of the table.

  1. In Figure 2,please add the disadvantages of MAB, GP and GE breeding.

     Authors’ reply: Figure 2 has been implemented according to reviewer’s suggestion.

Reviewer 2 Report

Major comments.

L.497. “The application of GP, which is based on the estimation of the breeding value of individuals, computed using only genotypic information [112], for the improvement of tree species is still in its infancy.”

This phrase is incorrect. First, GP (or GS) is based on the use of both genotypic and phenotypic information. “Here we attempted to estimate the effects of ⁓50,000 marker haplotypes simultaneously from a limited number of phenotypic records (Meuwissen et al., 2001 = Ref. 112).” “GS program for tree breeding encompasses essentially two stages. The first one involves the definition of a “training population” of individuals that are genotyped and phenotyped to develop and cross-validate predictive models” and “In GS breeding values ​​are predicted for each trait separately (i.e., a separate GEBV for each measured trait)” (Grattapaglia, 2017, Status and Perspectives of Genomic Selection in Forest Tree Breeding). In this regard, it is necessary to make changes to the text and Fig. 2. Secondly, at least 60 studies on GS on forest trees are already known (Lebedev et al., 2020, Forests, 11, 1190), including disease and pest resistance in Castanea, Fraxinus, Picea and Pinus. In addition, GS study for poplar has been reported, including resistance to rust (Pegard et al., 2020, Front. Pl. Sci, 11:581954).

L.502. “While simulation studies applied to woody perennials corroborate the advantage of GP for the simultaneous improvement of hundreds of minor-effect loci [114]”.

In fact, the GS estimates thousands and tens of thousands of loci.

Minor comments.

Fig. 1. In the caption to the figure, the species names should be italicized.

L.601. The abbreviation GS is listed but not in the text.

References are not formatted according to the rules of Plants.

Ref. 128 and 129 are missing in the text.

Author Response

Authors would like to thank the Reviewer for the critical assessment of the manuscript.

All Reviewer's comments have been considered and the manuscript has been modified according to the suggestions.

Authors’ replies to each observation from the Reviewer are reported point-by-point below.

We sincerely hope that our revised manuscript meets the standard of Plants and will be accepted for publication.

REVIEWER 2

Major comments.

L.497. “The application of GP, which is based on the estimation of the breeding value of individuals, computed using only genotypic information [112], for the improvement of tree species is still in its infancy.”

This phrase is incorrect. First, GP (or GS) is based on the use of both genotypic and phenotypic information. “Here we attempted to estimate the effects of ⁓50,000 marker haplotypes simultaneously from a limited number of phenotypic records (Meuwissen et al., 2001 = Ref. 112).” “GS program for tree breeding encompasses essentially two stages. The first one involves the definition of a “training population” of individuals that are genotyped and phenotyped to develop and cross-validate predictive models” and “In GS breeding values ​​are predicted for each trait separately (i.e., a separate GEBV for each measured trait)” (Grattapaglia, 2017, Status and Perspectives of Genomic Selection in Forest Tree Breeding). In this regard, it is necessary to make changes to the text and Fig. 2. Secondly, at least 60 studies on GS on forest trees are already known (Lebedev et al., 2020, Forests, 11, 1190), including disease and pest resistance in Castanea, Fraxinus, Picea and Pinus. In addition, GS study for poplar has been reported, including resistance to rust (Pegard et al., 2020, Front. Pl. Sci, 11:581954).

Authors’ replies: Here, the reviewer raises two main questions: 1) the fallacy of the sentence reported in lane 507 and 2) the widespread use of GP in tree breeding. As for the first question, we fully agree with the reviewer as the sentence reported in the manuscript is misleading. GP (or GS) aims to predict the breeding value of individuals using only genotypic information. Accordingly, we changed the sentence reported in lane 507 of the manuscript (The application of GP, which aims to estimate the breeding value of individuals using only genotypic information [112] ….)

For the second point (“GP is still in its infancy in tree breeding”), the reviewer highlights that there are more than 60 GP studies carried out in forest tree species, some of which were conducted in poplar. We agree with the reviewer that the sentence mentioned above is questionable, anyway considering the in poplar there is only one study to evaluate pros and cons of GP, the sentence was changed as follows: GP is still in its infancy in poplar breeding.

L.502. “While simulation studies applied to woody perennials corroborate the advantage of GP for the simultaneous improvement of hundreds of minor-effect loci [114]”.

In fact, the GS estimates thousands and tens of thousands of loci.

Authors’ reply: Obviously, this depends on the availability of markers for genotyping woody perennials, which has reached the figures pointed out by the reviewer.

Minor comments.

Fig. 1. In the caption to the figure, the species names should be italicized.

Authors’ reply: Fig. 1 caption has been modified according to reviewer’s suggestion.

L.601. The abbreviation GS is listed but not in the text.

Authors’ reply: The abbreviation GS has been removed in the list of abbreviations following reviewer’s comment.

References are not formatted according to the rules of Plants.

Authors’ reply: All the references have been modified according to the rules of Plants as suggested by the reviewer.

Ref. 128 and 129 are missing in the text.

Authors’ reply: Ref. 128 and 129 have been removed from the references according to reviewer’s comment.

English has been improved across the whole manuscript.

Sincerely,

Chiara Biselli

Round 2

Reviewer 2 Report

The manuscript has been improved and may be published in its present form.